# Applications of DNA-Functionalized Proteins

**DOI:** 10.3390/ijms222312911

**Published:** 2021-11-29

**Authors:** Zhaoqiu Gong, Yuanyuan Tang, Ningning Ma, Wenhong Cao, Yong Wang, Shuang Wang, Ye Tian

**Affiliations:** 1College of Engineering and Applied Sciences, State Key Laboratory of Analytical Chemistry for Life Science, Jiangsu Key Laboratory of Artificial Functional Materials, Chemistry and Biomedicine Innovation Center, Nanjing University, Nanjing 210023, China; mg20340019@smail.nju.edu.cn (Z.G.); mf20340057@smail.nju.edu.cn (Y.T.); dz1834021@smail.nju.edu.cn (N.M.); cwh@smail.nju.edu.cn (W.C.); dg20340046@smail.nju.edu.cn (Y.W.); 2Shenzhen Research Institute of Nanjing University, Shenzhen 518000, China; 3Institute of Marine Biomedicine, Shenzhen Polytechnic, Shenzhen 518055, China

**Keywords:** DNA–protein hybrid structure, DNA nanotechnology, cancer diagnosis, precise programming

## Abstract

As an important component that constitutes all the cells and tissues of the human body, protein is involved in most of the biological processes. Inspired by natural protein systems, considerable efforts covering many discipline fields were made to design artificial protein assemblies and put them into application in recent decades. The rapid development of structural DNA nanotechnology offers significant means for protein assemblies and promotes their application. Owing to the programmability, addressability and accurate recognition ability of DNA, many protein assemblies with unprecedented structures and improved functions have been successfully fabricated, consequently creating many brand-new researching fields. In this review, we briefly introduced the DNA-based protein assemblies, and highlighted the limitations in application process and corresponding strategies in four aspects, including biological catalysis, protein detection, biomedicine treatment and other applications.

## 1. Introduction

As an important component of organisms, protein is closely concerned with the life activities of almost all the creatures on the earth. Researching on protein is, somehow, to investigate the mystery of lives, and facilitate human beings’ better understanding of the world. Proteins in nature have extremely complex folding structures. Meanwhile, they can be combined with some biomacromolecules, such as DNA, RNA and lipids, to form intricate structures [1,2,3,4,5,6], which have shown great application prospects in biocatalysis, chemical detection, biomedicine, food research, etc. [7,8,9].

Specific connections built between RNA and protein can be applied in human disease diagnosis and treatment research. Nevertheless, the interaction between them will be affected by stress response and the cellular environment [10]. Genetic variation in the RNA–protein interaction site may interfere with the function of RNA and protein, and further hinder the susceptibility to human diseases [11]. Lipid–protein binding technology can be adopted to regulate cell functions and develop drug release systems [12]. However, since proteins are sensitive to long fragment insertion or organic solvent denaturation, it remains an urgent problem that how to form effective lipidation through chemical reactions. Due to the inherent hydrophobicity of lipids, the synthesis of lipid–protein conjugates shows poor performance in terms of the yield and selectivity of lipidation [13].

In comparison, DNA strands, owing to their good chemical and thermal stability, exhibit a broader application potential. DNA strands are often used as programmable motifs with controllable flexibility/rigidity and length, proper candidates for information storage [14,15,16,17] and non-toxic carriers for organisms [18]. Among these studies, DNA is especially valued and employed for its characteristics in two respects, including the function of DNA itself, such as aptamers with high affinity and specific binding properties, and the designability of DNA nanostructures. According to the Watson–Crick model, the reversible process of double-stranded DNA coupling and decoupling paves the way for precise programming. With the rise and development of DNA nanotechnology [19], DNA can be folded from a simple strand into any desired shape by design, such as a Holliday junction [19], double-crossover (DX) structure [20], octahedron structure [21], etc. In 2006, the emergence of DNA origami technology greatly promoted the designability and complexity of DNA nanostructures [22]. A variety of two-dimensional and three-dimensional DNA origami structures were constructed to further precisely organize inorganic and organic nanoparticles [23,24,25,26,27], among which guiding protein assembly is an important part. The spatial site selectivity and specificity of DNA nanostructure can be used for organizing protein and controllably manipulating the various properties of proteins, such as controlling growth conditions, regulating the growth environment of protein [28] and its size [29], which will effectively enhance protein functionality in biomedicine, detection and other aspects.

In this review, we will briefly introduce the research progress and applications of DNA-guided protein assembly in four respects, involving biological catalysis, protein detection, biomedicine treatment and other applications, and then estimate the opportunities and challenges in each branch.

## 2. Biological Catalysis

The precise assembly of enzymes in space is of great significance in biocatalysis. Due to its excellent addressability and designability, DNA can be used to precisely locate enzymes, and to reveal the action principle of enzymes in nature, which facilitates better research on the biochemical reactions of life. With the development of DNA nanotechnology, DNA nanoplatforms enable us to control precise position of enzymes and release of enzymes in a specific condition.

### 2.1. Enzyme Site Control

Combining enzymes with DNA can precisely control the position of the enzymes, distance and direction regulation, as well as the diffusion method, etc. [30,31,32]. As early as 2010, Numajiri et al. used a DNA origami nanoplane as a working platform for large-scale enzyme nanoarrays [33]. This square DNA origami for capturing protein can achieve precise control of sites at the nanometer level and the number of loaded enzymes at the same time.

Besides precise assembly, the combination of multiple enzymes is also vital in biological catalysis. Therefore, researchers focus on designing DNA structures to achieve more precise control of the site, and exploring how the difference in experimental parameters influences the effect of enzymes, in order to realize the comprehensive applications of different enzymes [34]. For example, in 2012, Fu et al. introduced a synthesis method to put a two-enzyme glucose oxidase (GOx) and horseradish peroxidase (HRP) cascade on a square DNA origami sheet (Figure 1a) [35]. For two adjacent enzymes GOx and HRP, their enzyme activity will be affected by their intermediate distance. When the distance is in the range of 65~20 nm, the enzyme activity increases with the distance decreasing. By shortening the distance between enzymes, the cascade reaction activity is enhanced due to the increased local molecular concentration. Conversely, when the distance is less than 20 nm, the enzyme activity presents a tendency to be positively correlated with the distance. The DNA origami vector in this article tends to be a template for anchoring enzymes. In 2016, they went a step further to realize the application of DNA template platform to precisely control the relative distance of the GOx–HRP complex enzymes (Figure 1b) [36].

### 2.2. Enzyme Encapsulation

In biological catalysis, it is a crucial issue to maintain the stability and activity of the enzyme. Research represents that the design of a proper DNA nanostructure can protect enzymes from erosion in cells without hindering the action of enzymes [38,39,40,41], which is important in maintaining the catalytic performance of enzymes. Further, through the structural design of DNA, the release of enzymes can be controlled to trigger under specific stimulating environments. For example, inspired by the cell compartment, Zhao and his colleagues constructed a DNA nanocage to assemble enzymes for promoting catalytic activity and stability (Figure 1c) [28]. It was found that DNA nanocage not only maintained the stability of enzymes, but also promoted catalytic activity, which meant DNA nanostructures could be regarded as an auxiliary carrier for enzyme regionalization and the regionalized enzymes could perform stronger activity. Similarly, in 2017, Grossi and his group fixed an enzyme in a closed cavity similar to a vault [42]. The sidewall of this vault-like structure has low permeability, and this property can be modified by subsequent design. This DNA vault restricts the reaction of hydrolase with the substrate when closed, while promoting their reaction when opened [31]. This DNA nanostructure can also immobilize hydrolytic enzymes and control their reaction in a targeted manner. Glucose oxidase and horseradish peroxidase were attached to two half-cages, respectively, and then approached when the two half-cages were assembled. This DNA nanocage can increase the turnover number of DNA nanocaged enzymes and protect enzymes from the degradation of proteases to ensure the durability of enzyme action. Linko et al. designed a tubular DNA nanostructure and loaded glucose oxidase (GOx) and horseradish peroxidase (HRP) on it [43]. Each piece of the tubular DNA nanostructure can be regarded as a component for different enzymes, and the enzyme cascade reaction can be realized effectively through assembly, which may be infinitely stacked in theory. The firm tubular outer wall could regionalize the enzymes and improve the enzymatic activity.

### 2.3. Enzyme Transport

After realizing the static and precise control over position, stoichiometric ratio and spacing, the researchers continue to treat DNA as a framework to explore some dynamic reaction processes. In the process of synergistic action of multiple enzymes, the substrate needs to be controlled in a reaction channel to prevent intermediate substances from being released into the system [44,45]. Accordingly, investigating enzyme transport methods becomes the key point. DNA can be designed to form swing arms, for its coupling and decoupling property, to mediate the substrate channel. Here, we cited some examples of DNA as an enzyme transfer station. Flexible double strands can be put in many structural designs. For example, Yan et al. constructed a DNA swing arm capable of transferring the position of enzymes in 2014 [46]. The enzyme is fixed on a DNA tile structure, so the distance between the enzyme sites can be strictly controlled. Due to the presence of the swing arm, the substrate molecule is allowed to be flexibly transferred between multiple sites, which greatly improves the efficiency of the enzyme. The flexible swing arm can not only increase the activity of the enzyme, but also provide an intermediate platform for enzyme transfer. This structure can be used as the transfer station between a variety of proteins, which means it has a universal application prospect. Furthermore, in 2018, Chen et al. also put a swing arm on a square DNA origami [47], and added a light-responsive cofactor to the carrier so that the DNA swing arm can reversibly switch the position of the substrate channel. The system can achieve both precise control of the site and reversible adjustment based on light stimuli.

Otherwise, the DNA framework template itself has certain advantages. As was mentioned in 2016 by Zhang et al., the pH surrounding DNA is lower than the pH of the original solution, as DNA is negatively charged, which optimized the environment for enzyme action [48]. The analysis of pH environment brought new ideas to the changes in DNA-guided enzymatic reaction conditions.

The interaction between neighbor enzymes in the enzyme-catalyzed reaction can also be affected by designed DNA. For example, Mukherjee et al. introduced a method of programming the function of streptokinase to control the spatial and temporal activities of enzymes (Figure 1d) [37]. The DNA-linked protease inhibitor was designed like a switch to regulate fiber activity, produced by a streptokinase (SK) and plasminogen (PG) combination. This SK-PG conjugation finally turned out to be effective in a complex whole blood-clot environment. DNA can also be used as a carrier to form a more complex enzyme nanofactory by embedding enzymes on a DNA origami nanoplatform [49]. In this system, DNA is a completely recyclable component which can be specifically modified to control the enzyme labeling. This mechanism is also expected to be applied in simple single-protein control and even multi-proteins binding regulation.

The flexibility of DNA nanotechnology in design enables us to create precise and controllable functional nanomaterials. The combination of DNA nanostructures and enzymes provides a feasible solution for site control, encapsulation and transport of enzymes. In general, DNA-mediated enzyme assembly plays a significant role in the enzyme catalysis field, and has great development potential and prospects.

## 3. Protein Detection

As the direct executor of life activities, protein participates in almost all processes of life. Abnormal protein expression is closely related to the occurrence of many diseases, and detection of many diseases can be dependent on abnormal protein levels. Thus, the detection of proteins is of vital importance in disease diagnosis. Protein detection is usually based on the binding between antibodies and proteins for recognition, using enzyme-linked immunosorbent assay (ELISA) and Western methods. The specific binding between antibodies and proteins can achieve highly sensitive and selective protein detection, but the application is limited by the high cost and time-consuming procedures. As early as 1990, Tuerk and Gold proposed that aptamers could be applied for protein detection [50]. In the following thirty years, considerable works have been developed from aptamer screening methods to detection methods [51]. Compared with antibodies, aptamers can also have a strong affinity with proteins [52,53], and the thermal stability and designability of DNA can fulfill the needs of more detection environments. Besides, with the development of DNA nanotechnology, the advantages of DNA nanostructures gradually attract public sight. New solutions have been proposed towards some problems in protein detection, such as detection of low-abundance protein detection and the spatial arrangement of antibodies.

### 3.1. Aptamers for Protein Detection

Aptamer is an oligonucleotide sequence that can be bound to the target molecule with high affinity and specificity, and it is repeatedly screened from the random oligonucleotide sequence library synthesized in vitro by using systematic evolution of ligands through the exponential enrichment (SELEX) technology.

The detection of the binding between aptamers and proteins is usually interrogated by fluorescence and electrochemical techniques. For example, the detection principle was based on the turn-off fluorescence when the free prostate-specific antigen (PSA) aptamer, modified by 6-carboxyfluorescein (6-FAM), bound with PSA. This method provided a quick prostate cancer test at relatively lower cost. Otherwise, researchers have put forward some designs to obtain clear detection signals. For example, Xing and colleagues raised a method in which a fluorescence cross-correlation signal is visible only when the aptamer probes–protein sandwiching structure formed (Figure 2a) [54], despite the fact that when using electrochemical techniques, aptamers are usually combined with metal or inorganic electrodes. For example, Lee and his colleagues found a method for quick prostate-cancer tests (Figure 2b) [55]. In 2018, Li and colleagues proposed an aptasensor using fluorescence detection (Figure 2c) [56]. This sensor based on graphene oxide was developed for the highly sensitive and selective determination of human cardiac troponin I (cTnI).

In terms of electrochemical methods, for instance, Souada et al. selected an aptamer that could bind to PSA and be fixed on a quinone-based conducting polymer, which maintained a smooth current (Figure 2d) [57]. When PSA was captured, apparent current drop would be detected, and the current would recover when the complementary DNA strands were added. By observing the current rise and drop, researchers can directly characterize the detection results. Additionally, Cui et al. immobilized a long-chain alpha-fetoprotein (AFP) aptamer on gold electrode, and then combined peptides on it [58], which is proven to work in amplifying the detection signal. AFP aptamers could be specifically bound with AFP, and peptides can prevent the adsorption of other non-specific proteins, which could be monitored by electrochemical methods. Zhou and colleagues developed a label-free electrochemical aptasensor to discriminate adenosine triphosphate (ATP) by the binding of AFP with its aptamer. In this work, thionin acted as the medium for electron translation to produce electrochemical signal for detection [59]. Similarly, Yang and colleagues proposed a label-free aptasensor based on graphene oxide (GO) that could actualize efficient and accurate detection towards AFP [60]. First, they cover the graphene oxide on the glassy carbon electrode (GCE) via chemical methods, then the NH_2_-funtionalized aptamer was bound to the carboxylated GO through the combination of carboxyl and amino groups. The introduction of carboxylated GO could improve the availability of aptamers and detection sensitivity. The combination of aptamer and AFP would block the electron transfer on the electrode; the decreased current signal was employed to quantify the AFP concentration. In addition, some other methods can be used to protein analysis and detection, including chemiluminescent detection [61,62,63], colorimetry detection [64,65], radioactive decay-based detection [66,67], etc.

### 3.2. DNA Nanostructures-Assisted Protein Detection

With the development of DNA nanotechnology, DNA can theoretically be designed and constructed into any shape. Due to the existence of steric hindrance, DNA nanostructures possess the possibility to resist nuclease attack and the capability to sustain their structural integrity for a long time. On basis of the mature DNA nanostructure, more and more research is being devoted to the application of DNA nanostructures for improving targeting and specificity of protein detection. For example, researchers found that the uncontrolled orientation of antibodies influenced the sensitivity of protein tests. Thus, in order to increase the sensitivity for prostate-specific antigen (PSA) detection, Zuo and colleagues used a DNA-based frame of similar size with PSA antibody to precisely guide the assembly of antibodies (Figure 3a) [68]. Importantly, the size-matching between DNA nanostructure and PSA antibody can minimize the steric effect and the overlapping of binding sites of antibodies, which is beneficial for improving the sensitivity. In addition, studies found that the advantage of DNA programmability could achieve controllable protein binding. In order to regulate the distance between the loaded antibodies, Yan et al. used DNA tile to assemble high-density aptamers microarray with controllable distance between aptamers [69]. The binding of thrombin with its aptamer arrays would replace the fluorescent nucleotide analogue, which allowed the detection of thrombin protein of low concentrations.

Currently, the detection of low-abundance protein biomarkers remains a challenge due to the low response signal and strong interference signal from the surrounding environment and other proteins. Therefore, it is necessary to improve the specificity and sensitivity. Komiyama and colleagues first designed a functional “DNA origami plier” to detect protein (Figure 3b) [70]. This “plier” consisted of two levers connected at a fulcrum where the specific protein was anchored. The opening and closing of the DNA origami plier was triggered by the interaction between target proteins, and then observed using AFM images at molecular resolution. Based on it, Walter and his colleagues used the same DNA origami to construct a traffic-light structure [72]. There were split aptamers separately on the two arms of plier, and the fluorescent color appeared green when the plier was opened. When ATP emerged, the two arms were closed due to the combination of the aptamer and ATP. A red fluorescent signal was thus detected because of the changed configuration and consequently energy transfer occurred. Moreover, DNA can be used to guide aptamer for detecting specific structural proteins. For example, because of the unique spatial structure of dengue virus (DENV), Kwon and colleagues constructed a pentagram-shaped DNA scaffold composed of tiles, with five molecular beacon-like motifs, which could be precisely aligned with the dengue envelope protein domain III (ED3) clusters on the surface of DENV (Figure 3c) [71]. The specific binding of DENV and its aptamer recovered the quenched fluorescence by separating the fluorophores from quenchers, which was used to realize the detection of DENV. This strategy can also be used to construct different DNA structures to detect viruses and antigens. All the above-mentioned examples proved that DNA provided an excellent platform to assemble proteins and execute protein functions. To sum up, DNA has irreplaceable advantages in protein detection, and more applications are expected to be developed.

## 4. Biomedicine Treatment

Lots of diseases arise from changes in proteins in cells [73], and protein therapy is a direct and safe method of disease treatment. To efficiently utilize protein drugs, two important issues need assuring, including safely delivering drugs to cells and controllably activating their efficacy. As the application of bare protein delivery is less efficient and poorly targeted, lots of materials have been employed to assist the delivery of protein drugs [74,75,76,77,78,79,80], among which the specific DNA-modified protein against tumor is a hot research object in the field of anticancer drug delivery. Using DNA with good biocompatibility as a platform to deliver drugs can effectively improve the drug release efficiency, the targeting and durability of the protein and the maximum tolerance of non-target cells for reducing the side effects.

### 4.1. Precise Drug Delivery

Due to the structural designability and ease of modification, DNA nanostructure is broadly viewed as potential candidate for precisely targeted delivering therapeutics, including protein drugs, into cells. In 2014, Liang et al. verified the controllability of DNA structure in targeted delivery through an interesting experiment. They designed tetrahedral DNA nanostructures (TDNs) as a carrier and investigated their trajectory in mammalian cells after endocytosis by single particle tracking method [81]. It was found that once TDNs entered the cell, they were regarded as foreign invaders and would enter the lysosome along the microtubules, which is a common phenomenon in drug delivery. Whereas, by decorating signaling peptides, TDNs can escape from lysosomes and be directed to the nucleus. Significantly, TDNs without NLSs were mostly restricted to the cytoplasm. The results strongly confirmed the effectiveness of protein-dressed DNA nanostructures as a delivery platform for targeted drug therapy.

Based on targeted delivery, it is another important issue to study the carrying capacity of drug carriers. For example, the key issue in developing new vaccines is how to efficiently activate the antigen-specific B cells. It has been reported that the direct linking between adjuvants and antigens could induce a strong B-cell response [82]. Hence, in order to maximize vaccine immunogenicity, molecular complexes need rationally designing. To this end, Liu et al. performed the first evidence on DNA nanostructure-based vaccine construction. They designed tetrahedral DNA nanostructures to carry a mixture of antigen (streptavidin) and CpG adjuvants, and to precisely control the valence and spatial arrangement of carried elements (Figure 4a) [83]. In vivo experiments showed that the DNA-guided complex can lead to a strong and long-lasting antibody response without stimulating a reaction to the DNA nanostructure itself, which has important implications for the efficacy and development of vaccines. Apart from linking between adjuvants and antigens, multivalent antigens can also induce positive effects on the activation of B cells [84]. Based on this, a lot of research was reported [85,86], but some specific factors that affect the activity of B cells by antigens remain uncertain. In 2020, Veneziano et al. conducted experiments to explore the influence of antigens’ quantity, spacing and the involved rigidity of scaffolds on the activation of B cells (Figure 4b) [87]. They designed and constructed a hollow icosahedron and a rod-like six-helix bundle DNA origami to connect germline-targeting (GT) engineered outer domain (eOD), the antibody for HIV-1 envelope glycoprotein antigen gp120, on its corresponding spatial programmed positions. Through choosing and changing different connection sites, they realized independent control over antigen stoichiometry, nanoscale spatial arrangement of antigens. Consequently, the design criteria that could maximize early B-cell triggering were eventually identified according to the above-mentioned investigation. In addition to the complex geometry of the structural design, DNA nanorobots, a kind of DNA origami with dynamic mechanical functions, were also designed and widely used in precise drug delivery [88,89]. Li et al. constructed a kind of DNA nanorobot for the transportation of thrombin (Figure 4c) [90]. This DNA–protein combination can be directly transported to the tumor-related blood vessels through intravenous injection of this thrombin nanorobot, and activate intravascular thrombosis and tumor necrosis without activating the immune system. The DNA–protein nanostructures greatly enhanced the targeting of drugs, and the evaluation results presented a significant improvement in antitumor effect and safety.

It is worth noting that DNA optimization for the treatment of protein diseases can not only be applied to single proteins, but also regulate multiple proteins to work together. For example, Zhu et al. constructed a combination of rolling circle replication (RCR) and rolling circle transcription (RCT) to generate intertwining DNA-RNA nanocapsules (iDR-NCs) in the same reaction system in 2019 [92]. IDR-NCs can realize the synergistic effect of carrier DNA and short hairpin RNA, and activate the sustained action of antigen. The iDR-NCs are expected to be used as a multi-drug carrier for tumor treatment. Similarly, Kim et al. constructed a DNA flower model to encapsulate a variety of biologically active proteins (Figure 4d) [91]. However, in this study the load of protein drugs is difficult to quantify. Basing on this, in 2019, Zhao et al. designed a square DNA origami platform as a carrier to quantitatively load cytotoxic protein ribonuclease. Through modifying the DNA origami platform with cancer cell-targeting aptamers, they accomplished targeted delivery of the protein drugs into the cytoplasm of tumor cells to realize its killing function (Figure 4e) [18]. They also proposed that a variety of functional proteins can be combined to achieve a synergistic effect between different proteins.

### 4.2. Controlled Drug Release

With the development of medical technology, the precise control of drug release under certain conditions has also become a research hot spot. Although great progress has been achieved in the field of precise drug delivery, due to the requirements for on-demand and fixed-point release of proteins, it is also necessary to design precise and controllable drug releases [93,94]. DNA nanorobots with dynamic mechanical functions play an important role in controlled drug release. In 2012, Douglas et al. developed a DNA nanorobot controlled by logic gates [88]. This nanostructure was designed as a hexagonal prism, which opened along the prismatic direction after receiving a response from the specific receptors on the cell surface to achieve its targeting effect. In addition, they also loaded multiple materials on DNA nanostructures and designed different logic gates. The release of antibodies can be controlled through different logic combinations. This ingenious structure enabled DNA nanorobots to carry out high-targeting and multiple drug delivery. Recently, a pH-responsive DNA nanodevice was developed through incorporating molecular adjuvants and antigen peptides into a tubular DNA nanostructure (Figure 5a) [95]. The low pH environment could open the nanostructure, and the nanodevice vaccine elicited a potent antigen-specific T-cell response.

### 4.3. Increased Biocompatibility

For drug research and development, how to reduce the side effects of drugs and increase the maximum tolerated dose of cells is also quite significant. Compared with other carrier materials, such as metal nanoparticle, DNA has a non-ignorable preponderance in reducing the auto immunogenicity of the immune system, as it is highly biocompatible [99,100]. For example, the researchers loaded doxorubicin on the DNA carrier and interacted with the target protein. The article published by Jiang et al. in 2012 also emphasized the enhanced effect of the DNA templates on the carrier (Figure 5b) [96]. In this paper, the DNA was folded into a two-dimensional triangle to load doxorubicin. This DNA origami can promote the internalization of doxorubicin, thereby enhancing the killing activity of doxorubicin-resistant MCF 7 cells. This non-toxic biological carrier can be slowly digested in the body, which shows its superiority in biocompatibility. In 2013, Zhu et al. constructed a simple aptamer-tethered DNA nanotrain structure to deliver drugs, which had the ability of enhancing the highest tolerated dose of non-target cells and showed excellent antitumor effects and less side-effects [101]. This simple structure containing few DNA base complementary pairs has a high production yield. It is simple to operate and easy to assemble, which is expected to become a relatively good diagnosis and treatment carrier.

In addition, the drug release kinetics can be reasonably controlled by setting the different shapes and structures of the DNA spiral twist [102]. A properly designed DNA framework can even play a positive role in promoting the function of drugs. In 2019, a new kind of nanorobot was reported for specifically inducing HER2 lysosomal to degrade membrane proteins [103]. The nanorobot was assembled by tetrahedral framework nucleic acid (tFNA) loaded on anti-HER2 aptamer (HApt). In vivo experiments confirmed that the presence of the tFNA structure can stabilize HApt, which explain the reason why this nanorobot can more efficiently promote HER2 for lysosomal degradation. The results showed the application possibilities of nanorobots as carriers for protecting aptamers and helping strengthen the function of proteins. In addition, more complex DNA nanostructures such as hollow tetrahedrons, octahedrons, cubes and other large-cavity structures have been developed, which undoubtedly promote the biochemical application of protein assemblies [24].

Although DNA–protein research is vigorously developing, some restrictions do exist in this industry, such as the economic cost of putting a large amount of DNA materials into practical use, and the balance between the cell uptake efficiency and cell uptake tolerance (Figure 5c) [97], which deserve further study and improvements in the future. Furthermore, the development of DNA platform, as a drug delivery tool, is encountering an obstacle as its low cell uptake rate. Schaffert et al. modified iron transport protein transferrin (Tf) on the DNA vector (Figure 5d) [98]. Compared with the unmodified structure, the cellular uptake rate of DNA loaded with Tf molecule modification increased 22 times. This ratio is positively related to the number of Tf on the surface. In order to solve the problem of weak stability and low infection rate of DNA origami, Auvinen et al. proposed another strategy [104], which utilizes the electrostatic interaction between protein–dendron conjugates and DNA origami to cover the entire DNA origami structure with protein. The results have proven that the coating structure can indeed improve the overall stability and transfection efficiency, and even protect the DNA origami structure from being discovered by the immune system.

In summary, as a drug delivery carrier, DNA nanostructures can realize targeted delivery and controlled release of protein drugs, and have excellent biological stability and compatibility, which provide an unparalleled platform for drug delivery. The clinical application of DNA nanostructures as drug delivery carriers will be one of the main goals of future research. Nevertheless, how to conveniently synthesize DNA structures in large-scale remains an urgent problem that needs to be solved.

## 5. Other Applications

In addition to the main applications described above, the DNA–protein composite structure can also be applied to other applications such cell nanoenvironment analysis and structure inspection. Analyzing the compositions and spatial organizations of the protein will benefit understanding of their functional relevance. For instance, Ambrosetti et al. utilized termed nanoscale deciphering of membrane protein nanodomains (NanoDeep) to analyze uneven proteins on the plasma membrane [105]. They designed a “nanocomb” structure to convert the Her2 membrane receptor protein organization information into DNA sequence information, which enabled us to know the nanoenvironments of a cancer-related receptor protein and offered new insights into the relationship between protein nanoenvironments and membrane protein functions. Cell nanoenvironments analysis is an important means for human beings to analyze the life process, because it can help monitor the complex nanoenvironments, which are composed of different biological molecules in space. Cell nanoenvironments monitoring mainly concerns how to detect the surrounding proteins in nanoenvironments. Chen’s group introduced a structure of macromolecules-tethered DNA walking indexing to probe the biological environment around multi-DNA-modified histone post-translational modification (PTM). They indicated that this one-to-many nanoenvironments detection was expected to be used in clinical applications [106]. Furthermore, DNA can also be applied for structure inspection so as to better understand life activities. In 2016, Funke et al. proposed a strategy to measure the force between nucleosomes using DNA origami [107]. They located the nucleosomes at the corresponding positions of the two DNA pillars. Under the action of the spring, the nucleosomes can attract each other to guide the two DNA pillars moving closer. Through the mechanical analysis of the entire structure, the Boltzmann-weighted distance-dependent energy landscape can be captured, the study of nucleosomes can help better understand the structure of cytoplasm.

## 6. Summary and Prospect

Proteins are essential components in the life activities of organisms, which have inspired researchers from multidisciplinary fields to simulate the protein assemblies in nature and understand the structure–function relationship. In this review, we concluded the typical progress in the application of DNA-based protein assembly. DNA with good biocompatibility and precise programmability has become a suitable tool to guide protein assembly. Large-scale DNA nanostructures, with high controllability and specificity in shape and sites, present more possibilities for protein assembly. For the application on protein detection as disease markers, the sensitivity and specificity were improved by introducing the functional DNA aptamer or controlling the orientation of proteins. Further, proteins as drugs are delivered into cells effectively and exerted higher efficacy by taking DNA as a carrier, which makes huge advances in biomedicine diagnosis and treatment. Furthermore, this review introduced studies on biological catalysis activity and nanoenvironment analysis for enzymes, which supplied more information to understand the internal therapeutic mechanism.

However, challenges existed in applications of DNA-guided protein assembly, for instance, high cost of DNA synthesis, tedious purification and limited reaction conditions for binding DNA and protein. Besides, given the behavior and function of natural protein assemblies, it deserves further investigation into the dynamic response of these artificial proteins. Finally, sufficiently understanding the relationship of structures and functions has been a challenging task. Considering the booming momentum of DNA nanotechnology-mediated protein assembly, more and more application fields, it is widely supposed, will be bound to establish in the near future. With precise design [108,109], DNA and proteins can be combined to form a more intricate protein network. For example, in an enzyme reaction, DNA can not only locate the enzyme site, but also control the microenvironment around the enzyme [110]. The combination of DNA technology and protein network in detection also has broad applications. With the help of high-resolution imaging technology, the interaction of protein–protein, protein–small molecule and protein–DNA can be characterized for early diagnosis of disease. Based on the demand for strict control over drug delivery in disease treatment, the application of DNA sensors may also have greater application prospects [111].

## Figures and Tables

**Figure 1 ijms-22-12911-f001:**
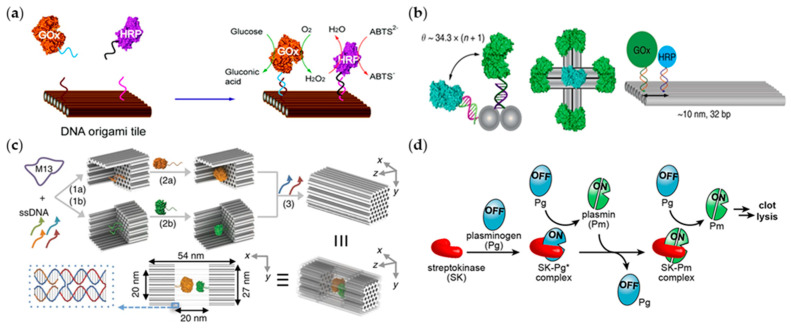
Enzymes combine with DNA to complete complex structures. (**a**) The way GOx and HRP work on DNA origami tile (adapted with permission from [35]. Copyright 2012, American Chemical Society.); (**b**) specific design of GOx and HRP enzymes integrated on DNA origami tile (adapted with permission from [36]. Copyright 2016, Springer Nature.); (**c**) encapsulate enzymes with DNA nanocages while molecules diffuse from the holes in the bottom of the cage (adapted with permission from [28]. Copyright 2016, Springer Nature); (**d**) the process of SK and PG forming a complex (adapted with permission from [37]. Copyright 2018, American Chemical Society).

**Figure 2 ijms-22-12911-f002:**
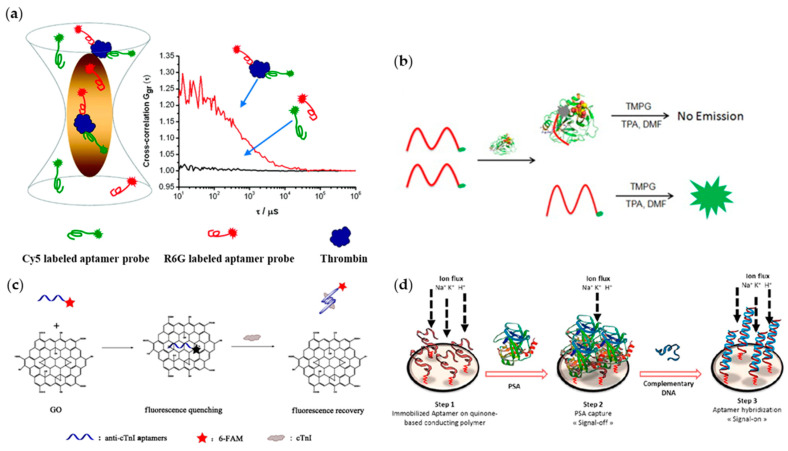
Protein detection based on DNA aptamers. (**a**) Aptamer probes-protein sandwich for fluorescence detection (adapted with permission from [54]. Copyright 2011, American Chemical Society); (**b**) one-step test system for PSA detection (adapted with permission from [55]. Copyright 2014, Elsevier); (**c**) a fluorescent aptasensor for detection of cTnI (adapted with permission from [56]. Copyright 2018, Springer Nature); (**d**) an aptamer probe for electrochemical detection (adapted with permission from [57]. Copyright 2015, Elsevier).

**Figure 3 ijms-22-12911-f003:**
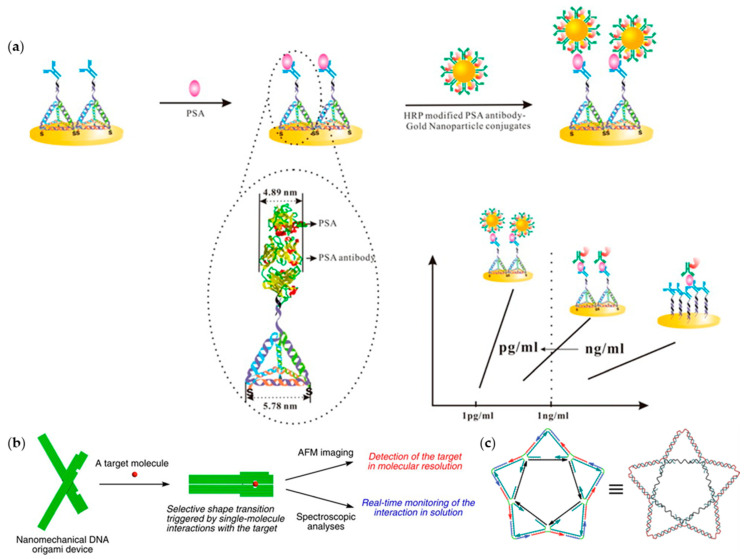
Protein detection based on DNA nanostructures. (**a**) DNA frame to control the assembly of antibodies (adapted with permission from [68]. Copyright 2014, American Chemical Society); (**b**) a functional “DNA origami plier” used as a protein test platform (adapted with permission from [70]. Copyright 2011, Springer Nature); (**c**) a pentagram-shaped DNA scaffold for DENV tests (Adapted with permission from [71]. Copyright 2019, Springer Nature).

**Figure 4 ijms-22-12911-f004:**
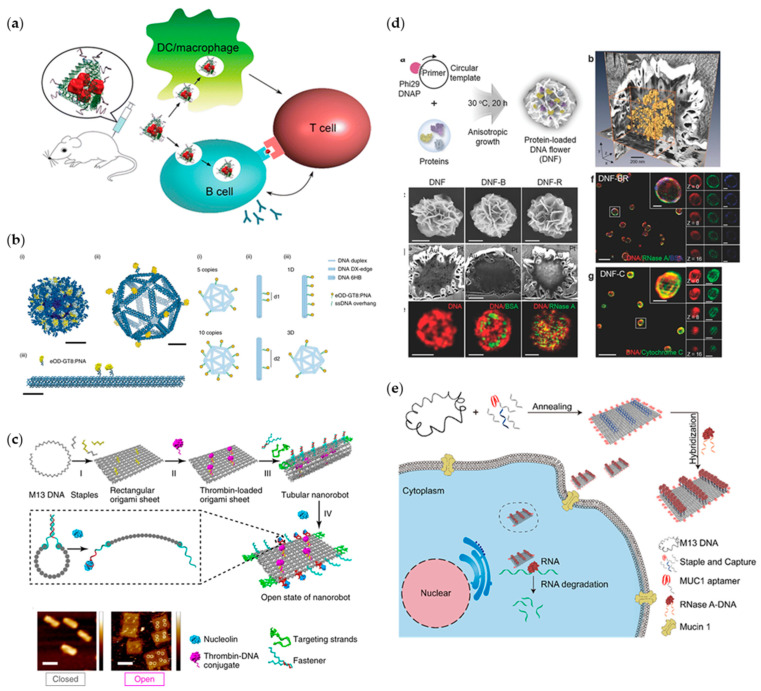
The schematic shows the design of programmable DNA structure as carriers. (**a**) DNA carrier platform loading streptavidin (adapted with permission from [83]. Copyright 2012, American Chemical Society); (**b**) DNA hollow icosahedrons for HIV immunogens (adapted with permission from [87]. Copyright 2020, Springer Nature); (**c**) A kind of DNA origami nanorobot (adapted with permission from [90]. Copyright 2018, Springer Nature); (**d**) DNA flower structure (adapted with permission from [91]. Copyright 2017, WILEY−VCH Verlag GmbH & Co. KGaA, Weinheim, Germany); (**e**) Rectangular DNA origami platform (adapted with permission from [18]. Copyright 2019, American Chemical Society).

**Figure 5 ijms-22-12911-f005:**
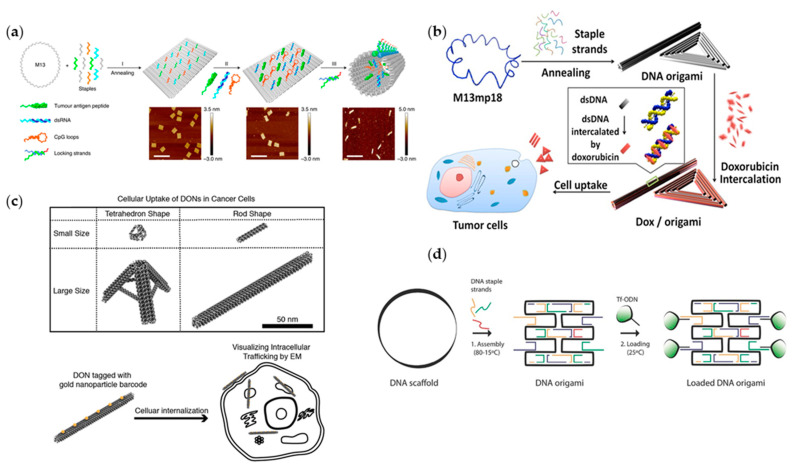
DNA–protein combination for complex application scenarios. (**a**) pH stimulus responsive DNA origami platform (adapted with permission from [95]. Copyright 2020, Springer Nature); (**b**) triangular DNA Origami Carrying Dox (adapted with permission from [96]. Copyright 2012, American Chemical Society); (**c**) DNA nanostructure for precise loading protein (adapted with permission from [97]. Copyright 2018, American Chemical Society); (**d**) DNA oligos carrying Tf (adapted with permission from [98]. Copyright 2016, WILEY-VCH Verlag GmbH & Co. KGaA, Weinheim).

## Data Availability

Not applicable.

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
