# Peer review of "Applications of DNA-Functionalized Proteins"

_ijms, 2021, doi:10.3390/ijms222312911_

Round 1
Reviewer 1 Report
This is a very interesting and well written review that I find worth publishing. As a minor comment: it would be interesting to add the few examples that exist involving peptide polymers (ring-opening polymerization). Cf for example the following articles : Chem Commun 2017, 53, 7501; Biomacromolecules 2018, 19, 4068 and Polymers 2020, 12, 1357.
Author Response
Responses to the reviewers:
Reviewer #1:
This is a very interesting and well written review that I find worth publishing. As a minor comment: it would be interesting to add the few examples that exist involving peptide polymers (ring-opening polymerization). Cf for example the following articles: Chem Commun 2017, 53, 7501; Biomacromolecules 2018, 19, 4068 and Polymers 2020, 12, 1357.
Reply: We thank the reviewer for the positive evaluation and highly appreciate for pointing out the important issue on peptide polymers. We have retrospected these representative works in appropriate parts and cited corresponding publications (ref. [80] [108] [109]) in the revised manuscript.
Reviewer 2 Report
In this review authors tried to summarize the current knowledge in the field of Assembly and Applications of DNA-functionalized Proteins. Truth is, that this title is slightly missleading, mainly the Assembly is not discussed to a sufficient detail.
The English in first part of introduction section is really poor and needs a lot of corrections and rephrasing (as well as in other parts of the manuscript).
The part about DNA origami is not in logical order.
In part about protein detection, authors selected an interesting work as an example, but didnt explain why this paper (was it first in the field? did it bring a new approach? did it shift the viewpoint? Also true further chapters). Also, even though there is a brief introduction in the part 2, I would recommend more general description of aptamers and methods of detection (on line 94 authors state "as mentioned above" but above there is just one example (that might be a general representation, but its not clear from the text; also true in other parts - authors should start with a paragraph of general principles and then explain them on selected examples).
Also the order of presented papers is chaotic (Not in chronological order, not exactly in methodological order - maybe subsectionig this chapter would make it easier to read (e.g. aptamers, DNA-origami, combination of aptamers with antibodies as subchapter).
Also authors should explain, why do they describe only one-step detection mechanisms, when using antibody (or aptamer) isolation (e.g. magnetic beads pull-down) is often used solution for the signal to noise ratio.
Simmilarly for part 3 - its chaotic. Dividing into subchapters would make it more readable and understandable. I would also suggest to go into greater detail.
If biological catalysis is one of the most common applications, maybe it would be wise to start with it and not have it on line 314.
Generally I am not sure, who is the expected reader of this review. For a person, who is working in this field it doesnt bring much new information and for someone who is not in the field the description of used methods and principles is to shallow. I would recommand to go into greater detail in explaining principles of described methods and also add a paragraph of some future perspectives.
Also 69 references are not much for a review article, which also shows that it is kind of shallow.
Author Response
Responses to the reviewers:
Reviewer #2:
In this review authors tried to summarize the current knowledge in the field of Assembly and Applications of DNA-functionalized Proteins. Truth is, that this title is slightly misleading, mainly the Assembly is not discussed to a sufficient detail.
Reply: We highly appreciate the reviewer’s professional comment about our manuscript and pointing out the problems. In this manuscript, emphasis was given to reviewing the typical progress in the application of DNA-functionalized proteins and thus the assembly methodology of DNA-protein complexes was shortly introduced We approved the comment given by this reviewer that the title is potentially misleading to readers. For the sake of clarity, we changed the title to "Applications of DNA-functionalized Proteins" for pointing out the keynote of this review and avoiding unnecessary misunderstandings. “Applications of DNA-functionalized Proteins”.
The English in first part of introduction section is really poor and needs a lot of corrections and rephrasing (as well as in other parts of the manuscript).
Reply: We thank the reviewer’s suggestion. We have made effort in the overhauling the language and writing of this manuscript and modified many unclear descriptions in previous version. We hope the revised manuscript will be easier for audience to understand.
The part about DNA origami is not in logical order.
Reply: We thank the reviewer for pointing out this critical issue.. We have revised this part by briefly describing the development of DNA nanostructures (including DNA origami) in chronological order, and citing/describing corresponding research papers to give readers more inspirations.
In part about protein detection, authors selected an interesting work as an example, but didnt explain why this paper (was it first in the field? did it bring a new approach? did it shift the viewpoint? Also true further chapters). Also, even though there is a brief introduction in the part 2, I would recommend more general description of aptamers and methods of detection (on line 94 authors state "as mentioned above" but above there is just one example (that might be a general representation, but its not clear from the text; also true in other parts - authors should start with a paragraph of general principles and then explain them on selected examples).
Reply: We thank the reviewer for the careful reviewing of manuscript and for the detailed expertise. We have rearranged and rewritten the ‘Protein detection’ part. Specifically, we divided this part into two subchapters, including “Aptamers for protein detection”, and “DNA nanostructures assisted protein detection”. After a full introduction of the importance of protein detection, we put forwards the descriptions of aptamer and DNA nanostructures in protein detection. In the “Aptamers for protein detection” part, we briefly described the aptamer firstly and further described two representative methods in detail, including fluorescence detection and electrochemical detection, with some typical examples in chronological order. On this basis, in the part of “DNA nanostructure assisted protein detection”, we briefly described the structural characteristics of DNA nanostructures and their important role in assisting aptamers protein detection through introducing a few typical examples. Similarly, in other parts, we have also made some logical adjustments, starting from the general principles, and further introducing some key issues and relevant examples. We consider the reviewer’s concern is fully addressed after we optimizing the article structure.
Also the order of presented papers is chaotic (Not in chronological order, not exactly in methodological order - maybe subsectionig this chapter would make it easier to read (e.g. aptamers, DNA-origami, combination of aptamers with antibodies as subchapter).
Reply: We highly appreciate the professional suggestion from this reviewer. We classified each chapter and overviewed the relevant researches of each category in chronological order. For example, in the part of “Protein detection”, we divided the whole chapter into two subchapters, including “Aptamers for protein detection”, “DNA nanostructures assisted protein detection” (including DNA origami). The relevant literatures in each section were summarized chronologically.
Similarly, we further subsected the other two parts, including “Biomedicine treatment” and “Biological catalysis”, and rearranged the order of related researches. We expect these alterations will enhance the logicality and readability of our manuscript.
Also authors should explain, why do they describe only one-step detection mechanisms, when using antibody (or aptamer) isolation (e.g. magnetic beads pull-down) is often used solution for the signal to noise ratio.
Reply: We thank again the reviewer for careful check of this manuscript. The manuscript (ref. [55] in revised version) mainly describes a rapid detection method rather than a one-step method, and we have corrected our statement in the article and placed it in the section of fluorescence detection.
Simmilarly for part 3 - its chaotic. Dividing into subchapters would make it more readable and understandable. I would also suggest to go into greater detail.
Reply: We accept this comment with appreciation.We divided the “Biomedicine treatment” part (part 3 in the original version and part 4 in the current version) into three subchapters: “Precise drug delivery”, “Controlled drug release”, and “Increased biocompatibility”. We gave a more comprehensive summary of the importance of DNA nanostructures in biomedicine, and a more detailed overview of the specific structural design of DNA nanostructures to meet the specific needs of biomedicine. We hope that better understandablity would be shown after the reconstruction of the above mentioned chapter.
If biological catalysis is one of the most common applications, maybe it would be wise to start with it and not have it on line 314.
Reply: We highly appreciate the suggestion from the reviewer. We rearranged the order of the chapters, and changed it to “Biological catalysis” (part 2), “Protein detection” (part 3), “Biomedicine treatment” (part 4), and “Other applications” (part 5).
Generally I am not sure, who is the expected reader of this review. For a person, who is working in this field it doesnt bring much new information and for someone who is not in the field the description of used methods and principles is to shallow. I would recommand to go into greater detail in explaining principles of described methods and also add a paragraph of some future perspectives.
Reply: This concern has been followed. We have revised the entire manuscript by including more information to attract the readers, including conducting a more in-depth overview on the basic principle and methods of each part, further analyzing the current existing problems, and giving more possible future applications in the prospect section. We believe the revised manuscript will provide implications to researchers from this filed and other interdisciplinary areas, and also appeal to general readers since the imaginations and artistry emerged in such methodology.
Also 69 references are not much for a review article, which also shows that it is kind of shallow.
Reply: We thank the reviewer to put forward this problem. After combining the modifications mentioned above, more representative publications were supplemented and we gave more detailed descriptions on these works, as suggested by the reviewer.
Round 2
Reviewer 2 Report
Authors have replied to all my comments and this provided version is (in my humble opinion) much better. I hope authors found my comments useful. In this state I recommand the paper for publication.